# Mechanically strong MXene/Kevlar nanofiber composite membranes as high-performance nanofluidic osmotic power generators

Zhen Zhang[1], Sheng Yang[1], Panpan Zhang [1], Jian Zhang [1], Guangbo Chen[1] & Xinliang Feng[1]

Two-dimensional nanofluidic channels are emerging candidates for capturing osmotic energy from salinity gradients. However, present two-dimensional nanofluidic architectures are generally constructed by simple stacking of pristine nanosheets with insufficient charge densities, and exhibit low-efficiency transport dynamics, consequently resulting in undesirable power densities ($<1\,W\,m^{-2}$). Here we demonstrate MXene/Kevlar nanofiber composite membranes as high-performance nanofluidic osmotic power generators. By mixing river water and sea water, the power density can achieve a value of approximately $4.1\,W\,m^{-2}$, outperforming the state-of-art membranes to the best of our knowledge. Experiments and theoretical calculations reveal that the correlation between surface charge of MXene and space charge brought by nanofibers plays a key role in modulating ion diffusion and can synergistically contribute to such a considerable energy conversion performance. This work highlights the promise in the coupling of surface charge and space charge in nanoconfinement for energy conversion driven by chemical potential gradients.

---

[1] Center for Advancing Electronics Dresden (Cfaed) and Department of Chemistry and Food Chemistry, Technische Universität Dresden, 01062 Dresden, Germany. Correspondence and requests for materials should be addressed to X.F. (email: xinliang.feng@tu-dresden.de)

The delicate ion transport in biological ion channels lays the foundation for almost all life processes[1]. As a functional mimic of living systems, bioinspired nanofluidic ion channels have been extensively investigated across scientific and engineering disciplines[2–5]. The ion transport in the nanofluidic channel differs significantly from that in bulk due to the unique nanoconfinement effect, which may find application in many areas, such as nanofiltration, energy conversion, DNA sequencing, and resistive pulse sensing[6–10]. For example, a one-dimensional (1D) nanofluidic channel system has been found to be able to capture the renewable osmotic energy available from the salinity difference between sea water and river water. In the past years, diverse 1D nanofluidic channels, such as single boron nitride nanotube and porous Janus ionic polymer membrane have been designed for osmotic energy harvesting[11–13]. However, the current 1D channel systems generally suffer from high mass transfer resistance and low pore density, resulting in limited power density and low large-scale viability.

Recently, the large-scale integration of nanofluidic channels has stepped into a completely new stage, termed two-dimensional (2D) nanofluidics, in which the water and ion transport are confined in the interstitial space between planar 2D materials[14]. Many materials such as graphene, vermiculite, and molybdenum disulfide have been explored for the construction of 2D nanofluidic channels with superior ion selectivity, ion rectification, and ion-gating properties[15–21]. Compared with 1D system, the 2D nanofluidic channel membrane offers great superiority, such as facile fabrication, efficient chemical modification, and accelerated mass transport, anticipating great potential in osmotic energy conversion, while the relevant investigations are rather limited[22–24]. The reported membranes are basically formed by simple stacking of pristine 2D materials (e.g. graphene and carbon nitride) with insufficient charge density and exhibit monotonic surface-charge-governed ion transport dynamics[25–30]. The achievable maximum power density is below 1 W m$^{-2}$, which is far less than the commercialization benchmark of 5 W m$^{-2}$ [31]. Therefore, to strive for an economically attractive power output, state-of-art 2D material system with advanced architecture and more efficient ion transport dynamics remains to be developed. To this end, an emerging class of 2D materials, transition metal carbide and nitride (MXene) that emerges as graphene analog, becomes an ideal choice. The surface of delaminated MXene nanosheets is hydrophilic and has abundant Lewis acid Ti sites and hydroxyl groups that can act as charged units, which can greatly facilitate the confined transport of water and ions. MXenes and their composites have been mainly investigated in electrochemical energy storage, such as supercapacitors due to their excellent electronic conductivity and high redox capacitance[32–35].

Herein, we demonstrate MXene-based composite membranes as high-performance nanofluidic osmotic power generators. The charged aramid nanofiber (ANF) derived from the commercial Kevlar yarns is selected as an intercalating and interlocking agent between MXene nanosheets[36]. The composite membrane exhibits charge-governed ion transport and shows good cation selectivity. By mixing artificial river water and sea water, the output power density can achieve ~3.7 W m$^{-2}$, which is the highest value ever reported in both 1D and 2D nanofluidic channel membrane systems. Replacing the artificial water with natural water resource can contribute to a substantially high power density about 4.1 W m$^{-2}$, much closer to the commercialization benchmark (~5 W m$^{-2}$). Additionally, benefitting from the introduction of ultra-strong structural unit ANF, the composite membrane also exhibits excellent mechanical strength and good stability, showing great application prospects. Both experiments and theoretical calculations reveal the synergetic effect of the surface charge of the MXene itself and the space charge brought by ANF is the key reason for such a considerable power output. Although high

surface charge densities have long been acknowledged to be able to largely enhance the nanofluidic energy conversion behavior[37], their correlation with space charge in nanoconfinement has not been investigated yet, either from an experimental or a theoretical point of view. Our results suggest that such a correlation plays a crucial role in modulating the ion diffusion process and is able to synergistically enhance the energy conversion performance.

## Results

**Fabrication of the MXene/nanofiber composite membrane.** The Ti$_3$C$_2$T$_x$ (MXene) was prepared by selectively etching of Al layer from Ti$_3$AlC$_2$ (MAX) using HF acid (Fig. 1a)[38]. The obtained multi-layer Ti$_3$C$_2$T$_x$ shows an accordion-like structure (Supplementary Fig. 1). After intercalation using organic alkali and subsequent mild sonication (see the "Methods" section), the multi-layer Ti$_3$C$_2$T$_x$ can be exfoliated into single-layer and few-layer nanosheets (Fig. 1b). Mean size of the as-prepared MXene nanosheets is ~700 nm (Supplementary Fig. 2). The etching process can introduce rich surface functional groups including oxide (–O–) and hydroxyl (–OH), as indicated by the high-resolution X-ray photoelectron spectroscopy (HRXPS) analysis (Supplementary Fig. 3). ANF is prepared from one of the strongest man-made ultrastrong materials, Kevlar yarns (Fig. 1c), which consist of aligned poly(paraphenylene terephthalamide) (PPTA) chains connected by intermolecular hydrogen bonds. Kevlar can be split chemically into numerous nanofibers through abstraction of protons from PPTA with saturated potassium hydroxide (KOH) in dimethyl sulfoxide (DMSO) (see the "Methods" section)[36]. As characterized by transmission electron microscope (TEM) measurement, the as-prepared red colloidal dispersion contains a large number of ANFs with length in micrometer scale and diameter in the range of 5–10 nm (Fig. 1d and Supplementary Fig. 4). HRXPS reveals the existence of C−N, C=O, and functional COOH groups on the surface of ANF (Supplementary Fig. 5 and Supplementary Table 1).

The obtained MXene and ANF can form a stable mixed suspension in DMSO solvent with strong Tyndall Effect, indicating its good dispersibility (Fig. 1e). The MXene/ANF can be readily reassembled by vacuum-assisted filtration method to form a black gray, paper-like flexible thin membrane (Fig. 1f) with typical lamellar microstructure as shown in the cross-sectional scanning electron microscope (SEM) image (Fig. 1g). Here, the ANFs act not only as an intercalating agent to enlarge the interlayer spacing, but also as interlocking agent to connect the nanosheets by hydrogen bonding (Fig. 1h). The as-prepared composite membrane is hydrophilic with a contact angle of about 45° (Supplementary Fig. 6) and is electrically conductive with a sheet resistance about 37 kΩ sq$^{-1}$. The thickness is in the range from 2 to 15 μm which can be adjusted through the dosage of the colloidal dispersion used for filtration.

**Charge-governed ionic transport.** The transmembrane ionic transport properties of the composite membrane with a thickness of 4.5 μm were examined by ionic current−voltage (I−V) measurements performed using an electrochemical cell (Fig. 2a). Figure 2b shows a series of I−V curves recorded in KCl electrolyte with concentrations ranging from 0.05 to 1 M. They all behave as linear ohmic behavior with negligible ionic current rectifying phenomenon, which can be ascribed to the symmetric structure of the composite system. Conductance measurement reveals a charge-governed ionic transport through the composite membrane[39,40]. As shown in Fig. 2c, the measured ionic conductance has two distinct characteristic behaviors. In detail, the ionic conductivity follows the bulk rule with a linear relationship in high concentration region. While, when the concentration is

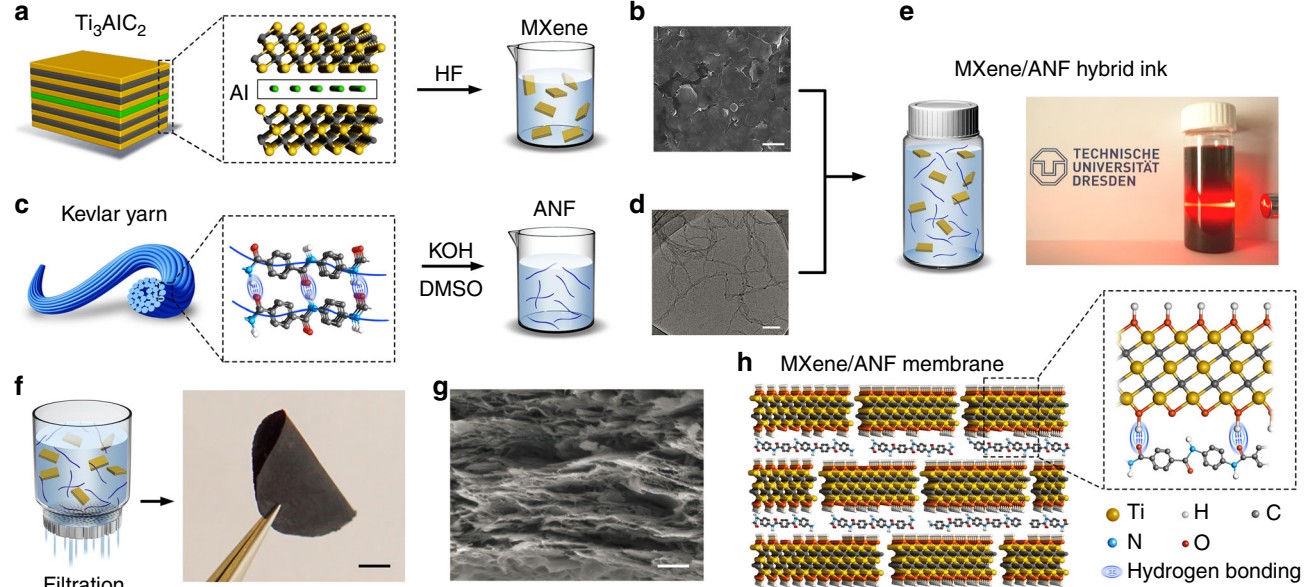

**Fig. 1** Fabrication of the MXene/nanofiber composite membrane. **a** The MXene is prepared from the $Ti_3AlC_2$ precursors using HF etching method. **b** SEM image of the as-prepared MXene nanosheets (scale bar: 1 μm). **c** The ANF is prepared from commercial Kevlar yarns using the KOH/DMSO exfoliation method. **d** TEM image of the as-prepared ANF (scale bar: 200 nm). **e** Suspension of ANF/MXene composite exhibits strong Tyndall effect. **f** A flexible, free-standing, paper-like MXene/ANF composite membrane with an ANF weight content of 11% obtained by vacuum-assisted filtration (scale bar: 5 mm). **g** Cross-section SEM image shows the lamellar microstructure of the composite membrane (scale bar: 400 nm). **h** Schematic of the interior structure of the MXene/ANF composite membrane

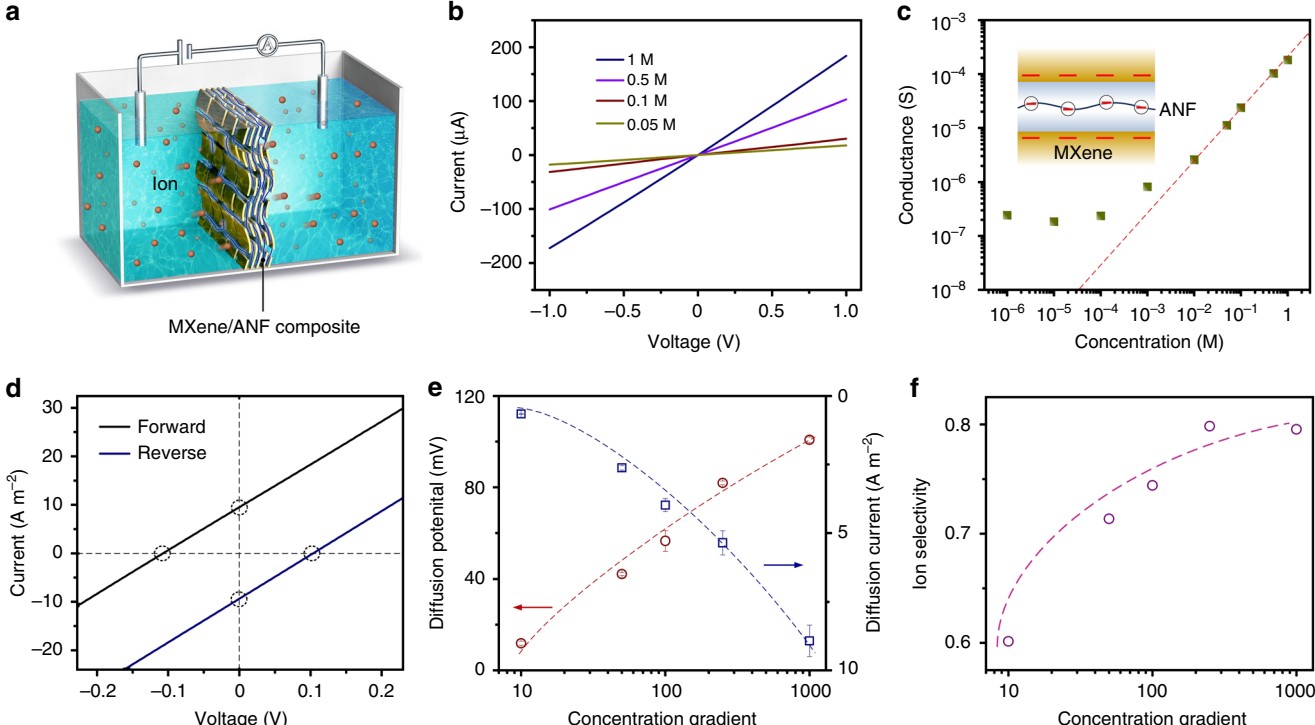

**Fig. 2** Transmembrane ionic transport. **a** Schematic of the experimental setup to measure the transmembrane ionic transport. **b** I–V curves of composite membrane with an ANF weight content of 11% recorded in neutral KCl electrolyte with different concentrations. **c** Conductance measurement in KCl electrolyte. The ionic conductance deviates from bulk value (red line) at low concentration region, indicating a charge-governed ion transport. Inset: The charged state of MXene and ANF. **d** I–V curves of composite membrane with an ANF weight content of 11% recorded in 1000-fold KCl concentration gradient under forward and reverse diffusion directions. **e** The recorded diffusion potential and diffusion current under a series of concentration gradient. The low concentration side is set to 0.1 mM. Error bars represent s.d. **f** The ion selectivity, defined as transmembrane ion transference number, increases from about 0.6 to 0.8 with increasing concentration gradient

<10 mM, the conductance deviates from bulk value and gradually approaches a plateau[41]. In low concentration region, the thickness of electric double layer increases and becomes comparable with the channel dimension. As the MXene and ANF both carry negative surface charge, cation (i.e. $K^+$) with the opposite charge will enrich in the channel and anion (i.e. $Cl^-$) with the same charge will be excluded, forming a confined unipolar environment. In this condition, the ion concentration inside the channel is determined by the charge on the MXene and ANF rather than the bulk concentration, and the increased ionic carrier concentration will contribute to the enhanced conductance[42]. Interestingly, besides $Li^+$, $Na^+$, and $K^+$ ions, the membrane also shows large conductance towards $Mg^{2+}$ ion, which is generally considered slow to migrate across the MXene membrane due to its large hydrated radius (Supplementary Fig. 7)[43]. In our case, the intercalation of ANF can enlarge the interlayer space between MXene flakes and prevent their restacking, thus resulting in a high $Mg^{2+}$ flux.

**Concentration gradient-driven ionic transport**. Next, chemical potential gradient by using the KCl concentration gradient system was applied across the composite membrane, which could provide further insight into the confined ion transport process. As the negatively charged composite membrane is cation-selective, it can transport cation (i.e. $K^+$) preferentially from high concentration side to low concentration side, generating the so-called diffusion current ($I_{diff}$) and diffusion potential ($V_{diff}$)[44]. By collecting $I–V$ curves in the presence of a transmembrane concentration gradient, the values of $V_{diff}$ and $I_{diff}$ can be directly derived from the measured open-circuit voltage and short-circuit current, respectively. Figure 2d shows the representative $I–V$ curves of the composite membrane (ANF content ~11%) recorded under 0.1 mM/10 mM KCl concentration gradient. The $I_{diff}$ and $V_{diff}$ are about 9 A m$^{-2}$ and 103 mV, respectively. Notably, the contribution of redox potential that is generated by the unequal potential drop at the electrode–solution interface has been subtracted via an electrode calibration process (Supplementary Fig. 8 and Supplementary Note 1). Under a reverse concentration gradient, the $I_{diff}$ and $V_{diff}$ show the similar values, but are of the different polarity due to the opposite ion diffusion, indicating that the membrane has a symmetric structure with no preferential direction for ion diffusion. Under a series of concentration gradient, the $I_{diff}$ and $V_{diff}$ both increase as the concentration gradient increases (Fig. 2e). Actually, the $V_{diff}$ originates from the ion selectivity of the membrane which result in the differences in the diffusive fluxes of anions and cations[45]. The $V_{diff}$ can be described as

$$V_{diff} = (t_+ - t_-)\frac{RT}{F}\ln\left[\frac{a_{high}}{a_{low}}\right]$$

Here $t_+$ and $t_-$ are the transference numbers for positively and negatively charged ions, respectively. $R$, $T$, $F$ are the universal gas constant, the absolute temperature, and the Faraday constant, respectively. $a_{high}$ and $a_{low}$ are the activities of KCl in the high concentration and low concentration side. By fitting the experimental data in Fig. 2e to the equation, $t_+$ and $t_-$ under a series of concentration gradient can be obtained. As shown in Fig. 2f, the ion selectivity that can be quantified by the transference number of major carriers (in this case the $K^+$) can approach about 0.8. As the interlayer spacing between MXene nanosheets is largely expanded by ANF, the achieved ion selectivity is quite considerable, comparable with that of the ultrathin single crystal $MoS_2$ nanopore system[30].

**Osmotic energy conversion behavior**. The harvested osmotic energy can be output to external circuit to supply an external load ($R_L$) (Fig. 3a). To evaluate the application prospects of the composite membrane system, standard artificial sea water (0.5 M NaCl) and river water (0.01 M NaCl) system was selected. Figure 3b shows the power generation of a composite membrane with an ANF weight content of 11%. As the resistance increases, the current density on the external circuit decreases accordingly, while the output power density, calculated as $P = I^2 \times R_L$, reaches a maximum value ~3.7 W m$^{-2}$ at an intermediate resistance of 27 kΩ, which is close to the internal resistance of the composite of about 29 kΩ (Supplementary Fig. 9). This power density is the highest value ever reported, exceeding all the reported 1D and 2D nanofluidic channel membrane systems to date. The corresponding energy conversion efficiency can approach ~35% (Supplementary Note 2), which is comparable with the commercial ion exchange membranes[46]. Further decreasing the membrane thickness to 2 µm can contribute to an increased power density of 3.92 W m$^{-2}$ (Supplementary Fig. 10).

The weight content of ANF inside the composite membrane has a significant influence on the power density. As shown in Fig. 3c, the power density first increases from 2.6 to 3.0 W m$^{-2}$, and then achieves a maximum value of about 3.7 W m$^{-2}$ upon increasing the ANF content. This is because the intercalation of ANF can enlarge the interlayer space between MXene flakes and contribute to enhanced ion flux through the 2D nanochannels (Fig. 3d and Supplementary Fig. 11). However, continuously increasing the ANF content will eventually undermine the capability of osmotic energy conversion. The power density drops substantially from 3.7 to 1.9 W m$^{-2}$ as the ANF content increases from 11% to 30%. In this respect, the excess ANF can partially block the 2D channel and introduce large physical steric hindrance for the ion transport, resulting in decreased ion flux and thus largely weakened power output. We also test the power output using other typical types of chloride electrolyte: LiCl with a power density of about 1.9 W m$^{-2}$ and KCl with a power density of about 4.6 W m$^{-2}$ (Supplementary Fig. 12). For the three types of chloride electrolytes, the diffusion coefficient of $K^+$ is the largest and the diffusion coefficient of $Li^+$ is the smallest[47]. As the system is cation-selective, the faster the cation diffuses, the more remarkable the charge separation will occur[48]. Therefore, the energy conversion performance of KCl electrolyte is the most remarkable one.

**Synergetic effect of space charge and surface charge**. Obviously, the composite membrane exhibits outstanding osmotic energy conversion capability. Here, the ANF can serve as an intercalating agent that will enlarge the interlayer channel and prevent the restacking of adjacent MXene nanosheets. More importantly, the ANFs are negatively charged in solution and their entanglement in confined channel can create a negatively charged space charge zone[49]. The coexistence of the surface charge of the MXene itself and the space charge brought by ANF will make the electric double layer confined in the nanochannel completely overlap. This synergetic effect will largely increase the ion selectivity and enhance the ion flux of the composite system, contributing to substantially high power density and conversion efficiency.

For comparison, MXene composite membranes in which neutral polyvinyl alcohol (PVA) serves as the intercalating agent were also fabricated. Figure 4a shows the energy conversion behavior of MXene/PVA composite membrane with the same weight content (~11%). Obviously, the power density of the MXene/PVA system could only reach ~2.2 W m$^{-2}$, which is significantly lower than that of the MXene/ANF system. This result indicates that the charge density of the intercalating fiber

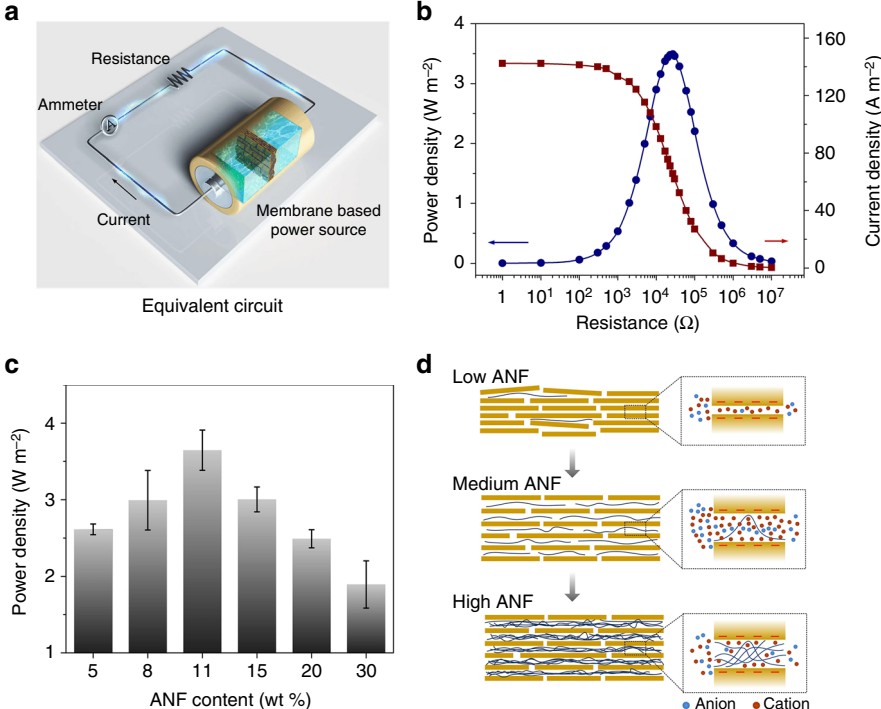

**Fig. 3** Osmotic energy conversion behavior. **a** The collected power can be transferred to the external circuit to supply an electronic load. **b** Power generation of a composite membrane with an ANF content of 11% by mixing artificial river water (0.01 M NaCl) and sea water (0.5 M NaCl). The current density decreases with increasing resistance, while the power density achieves a maximum value of ~3.7 W/m² at an intermediate resistance. **c** Influence of the weight content of ANF on the output power density. Error bars represent s.d. **d** Schematic of the influence of the ANF content on the energy conversion process

plays an important role in the power generation performance of the composite membrane (Fig. 4b). For composite membrane system with optimized fiber content (i.e. 11%), the enhancing effect of charge factor will even screen the blocking effect of physical steric hindrance factor. The space charge and surface charge synergistically governing energy conversion process can also be strongly influenced by the pH value of the electrolyte. As the pH increases from 4 to 10, the energy density increases from ~1.9 to 4.5 W m$^{-2}$ (Fig. 4c). For these electrolyte solutions, the proton concentration in water can be neglected. The enhanced power generation behavior can be ascribed to the variation of charge density of the MXene and ANF. As reflected by the Zeta electric potential measurements, both the charge density of MXene and ANF are pH sensitive (Fig. 4d) and can be strengthened upon raising the pH.

**Continuum-based theoretical simulation.** To provide further physical insight into the synergetic effect of space charge and surface charge, continuum-based theoretical simulation using Poisson and Nernst–Planck (PNP) theory was carried out[50]. The fluidic transport route inside the composite membrane is simplified to be a 2D single nanochannel (Fig. 5a, Supplementary Fig. 13, and Supplementary Note 3). As the interlayer spacings are largely expanded by ANFs, the size of the channel is set to 10 nm, consistent with the experimentally measured diameter of the nanofiber. The surface charge and space charge are set within reasonable ranges—in the same order of magnitude with the charge density of metal oxide/carbide[51] and polymer-brushes-filled nanochannel[52], respectively. As shown in Fig. 5b, the calculated power output of the composite system (model-3) increases accordingly upon either solely or simultaneously increasing the surface charge density and space charge density. Obviously, the power output is much higher than the

corresponding mathematical sum of the value in which single type of charge exists (i.e. model-1 and model-2) (Fig. 5c). This phenomenon can be ascribed to the synergetic effect of surface charge and space charge. The cumulative efficiency is greater than the sum of each type of charge used in isolation, which is owing to the exponentially increased electric potential when the electric double layers confined in the channel completely overlap[48]. The synergetic effect can contribute to a membrane potential of about 88 mV corresponding to an energy conversion efficiency of about 38%, comparable with the experimentally measured value (84 mV corresponding to efficiency 35%).

The synergetic effect originates from the correlation of surface charge and space charge in nanoconfinement and can be quantified by the enhancing factor ($R_E$) that is defined as the ratio of the power increase of model 3 and the corresponding mathematical sum of model-1 and model-2 ($\sigma$ and $\rho$ represent the surface charge density and space charge density, respectively):

$$R_E = \frac{P_{\text{Model}-3,\sigma,\rho} - (P_{\text{Model}-1,\sigma} + P_{\text{Model}-2,\rho})}{P_{\text{Model}-1,\sigma} + P_{\text{Model}-2,\rho}}$$

We further calculated the influence of the surface charge density, space charge density, and the interlayer distance on the synergetic effect-induced $R_E$. As shown in Fig. 5d, the relationship between $R_E$ and the surface charge density presents an unimodal distribution and $R_E$ achieves a maximum value of about 0.8 at an intermediate surface charge density ($\sim -0.02$ C m$^{-2}$). Similarly, it is also the case of the relationship between $R_E$ and the space charge density (Fig. 5e), where $R_E$ reaches a maximum value of ~0.65 at an intermediate space charge density of about $-8$ C cm$^{-3}$. These mean that such a correlation becomes most remarkable when the effect of surface charge on the ion transport is comparable with that of space charge. If one side is very powerful

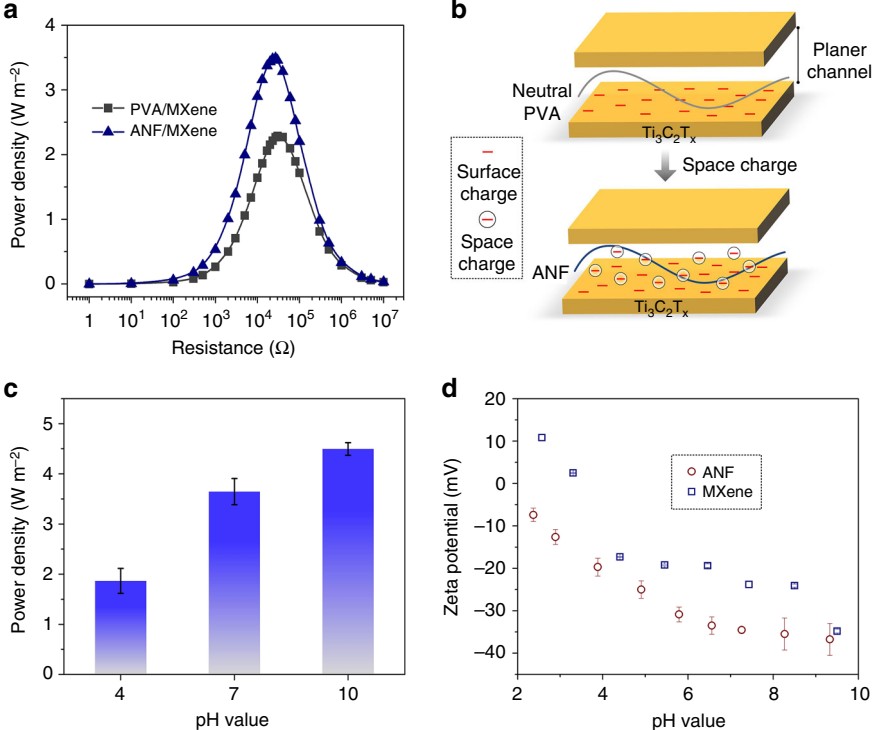

**Fig. 4** Synergistic effect of surface charge and space charge. **a** Comparison with the power output of MXene/PVA composite membranes with the same weight ratio as MXene/ANF composite. The power output of MXene/ANF composite membranes is plotted for a direct comparison. **b** Schematic of the synergistic effect of surface charge and space charge. **c** Dependence of the output power density on the electrolyte pH value. Error bars represent s.d. **d** Surface Zeta electric potential of the MXene and ANF as a function of pH value, indicating that their charge effect can be largely strengthened upon raising the pH. Error bars represent s.d

and one side is very weak, the synergistic effect will be undermined. The synergistic effect is also dependent on the interlayer distance of the channel and achieves a maximum value with a channel size in the range of 10–14 nm (Fig. 5f).

**Application viability of the composite membrane.** Membrane materials suitable for practical applications should have enough mechanical strength. Figure 6a shows the tensile stress–strain curves of the composite membranes (2 mm × 8 mm) with different ANF contents. As the ANF content increases, the strength and strain (i.e. the force and elongation at break) both increase accordingly. For the optimized membrane with an ANF content of 11%, the ultimate strength and toughness are ~101 MPa and 2.64 MJ m$^{-3}$, which is comparable with natural nacre[53] (i.e. about 130 MPa and 1.9 MJ m$^{-3}$, respectively) (Fig. 6b, c). The Young's modulus also achieves a considerably high value of about 3.4 GPa (Supplementary Fig. 14). The observed mechanical strength largely outperforms most of the reported membranes based on stacked pristine 2D materials[54], which can be ascribed to the introduction of ultrastrong structural unit ANF. Here, the MXene nanosheets and ANFs form a "brick-and-mortar" structure where the hard MXenes act as "brick" and the soft ANFs act as "mortar", similar to the structure of natural nacre[53]. The ANFs actually serve as interlocking agents to connect the MXene flakes by strong hydrogen bonding (Fig. 1h), resulting in such a high mechanical strength.

Besides pure electrolyte solutions composed of simple inorganic salts, the composite membrane is also capable of harvesting osmotic energy from natural water source, where sea water is obtained from *Mediterranean Sea* (~0.6 M NaCl) and river water (~0.004 M NaCl) is obtained from *Elbe River*. As shown in Fig. 6d, the maximum power density can achieve an amazing value of ~4.1 W m$^{-2}$, which

is about twice of the value of the state-of-art membranes[11] working in the similar natural water resource condition, and much closer to the commercialization benchmark (~5 W m$^{-2}$)[31]. With no continuous electrolyte supplying and replenishing, the current density on the external circuit only exhibits a 2.3% attenuation in the first six hours (Fig. 6e), implying that the membrane can effectively stabilize transmembrane concentration gradient and promote continuous osmotic energy harvesting. In addition, the system also shows excellent working stability. The measured power density and electric potential do not show any attenuation after several days. Even after one month, the composite membrane still maintains the power output (Fig. 6f and Supplementary Figs. 15, 16), which can be ascribed to the good chemical stability of the constituent material (i.e. ANF and MXene) and the interlocking mechanism. After the long-term stability test, the membrane was still very robust and there were no observable morphology changes. These observations clearly indicate the great application viability of the composite membrane.

## Discussion

In summary, we have explored the application of MXene-based composite membrane in osmotic energy conversion. The maximum power density can achieve ~4.1 W m$^{-2}$ for the natural sea/river water system, approximately twice of the value of the state-of-art membranes and much closer to the commercialization benchmark (~5 W m$^{-2}$). It should be noted that the testing membrane area was about $3 \times 10^4$ μm$^2$, the same as previous reports[55]. Normalizing high power density into real high power for industrial large-scale membrane applications remains challenging. Both experimentally and theoretically, we have revealed and highlighted the role of the nanoscale correlation of the surface charge of MXene and the space charge brought by ANF in synergistically improving the energy-conversion performance.

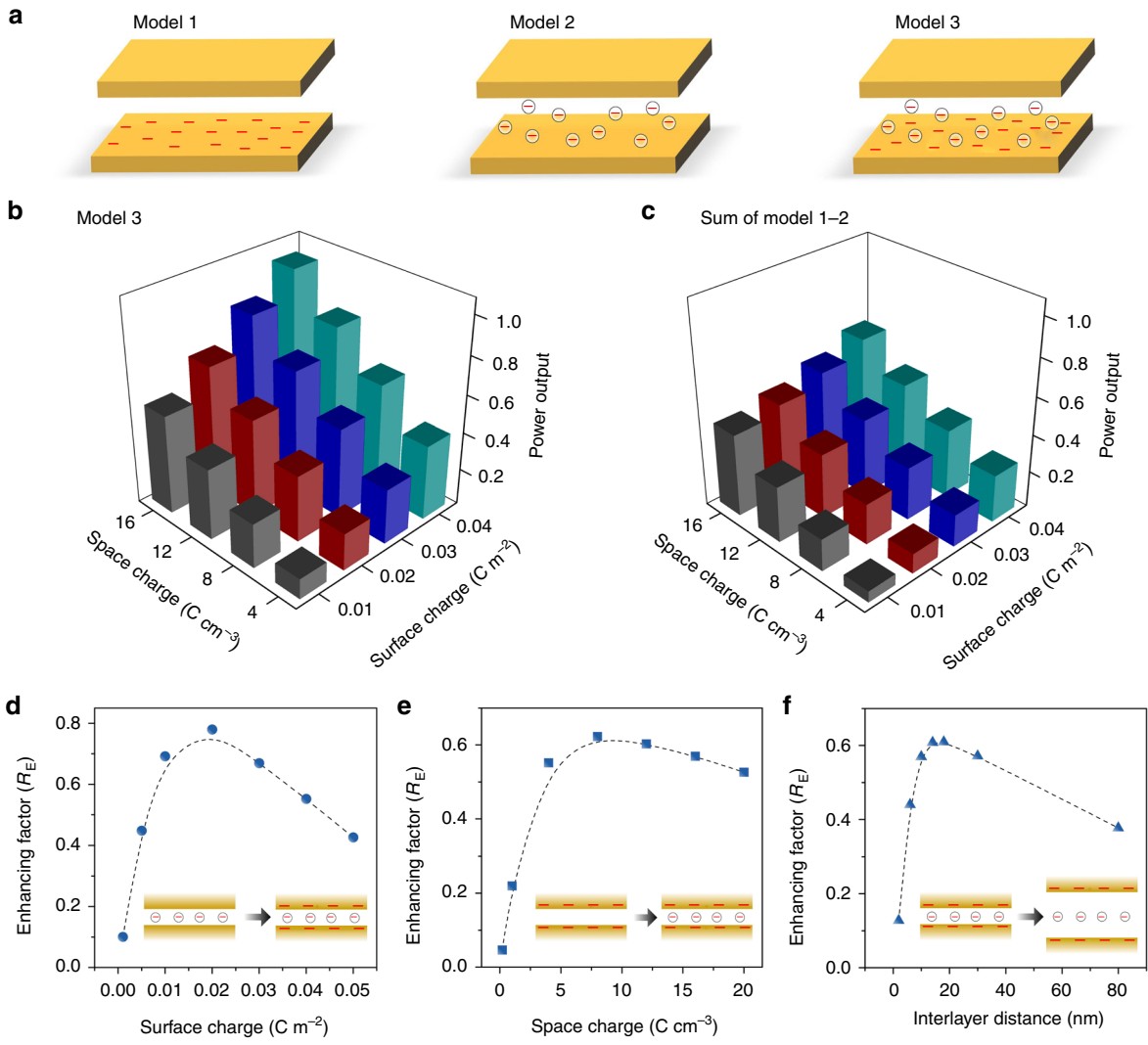

**Fig. 5** Continuum-based theoretical simulation. **a** Three theoretical models including solely surface-charged 2D channel (model-1), solely spacing charged 2D channel (model-2), and simultaneously surface and space charged 2D channel (model-3). **b** The calculated normalized power output by the model-3. **c** The calculated normalized power output by the mathematic sum of model-1 and model-2, which is much lower than the value of the model-3 for any charge combinations, indicating the remarkable synergetic effect of surface charge and space charge. **d** Influence of the surface charge on the enhancing factor when the space charge is set to be constant ($-4\,C\,cm^{-3}$). **e** Influence of the space charge on the enhancing factor when the surface charge is set to be constant ($-0.04\,C\,m^{-2}$). **f** Relationship between the interlayer distance of 2D channel and the enhancing factor. The charge densities are all negative values

Furthermore, our composite membrane also has excellent mechanical strength and outstanding stability, thus satisfying some requirements for practical applications. In this work, the mechanism of strongly enhanced nanoconfined ion diffusion by the synergistic effect of surface charge and space charge will enable next generation of advanced materials for osmotic energy harvesting. It may also open the route for the applications of such membranes in other membrane-based chemical potential gradient-driven energy conversion technologies, such as photo-electric, thermoelectric, and hydraulic energy conversion[56–58].

## Methods

**Materials**. Ti$_3$AlC$_2$ (200 mesh, purity >98 wt%) was purchased from Forsman Scientific Co., Ltd. Anodized aluminum filter film (pore size ~0.2 μm) was provided by Whatman. Kevlar yarns were purchased from Dupont. Tetramethylammonium hydroxide (TMAOH) and PVA ($M_w$~100,000) were purchased from Sigma. All the chemicals including hydrogen fluoride (HF), DMSO, KOH, sodium chloride (NaCl), potassium chloride (KCl), magnesium chloride (MgCl$_2$), and lithium chloride (LiCl) were analytically pure.

**Synthesis of MXene**. The MXene was synthesized based on the reported HF etching method[38]. In detail, 1 g MAX precursors were immersed in 40 wt% HF aqueous solution (10 mL) and the mixture was magnetically stirred for 24 h. The acidic product was washed with deionized water via centrifugation method until the pH of the solution was above 6.0. The obtained multi-layer Ti$_3$C$_2$T$_x$ was further intercalated using TMAOH with magnetically stirring for 6 h. The product was copiously washed and then dispersed in deionized water, followed by mild ultra-sonication. After 20 min centrifugation at 3500 rpm, the MXene colloidal solution could be obtained.

**Synthesis of ANF**. One gram of bulk Kevlar and 1 g KOH were added into 100 mL of DMSO. The mixture was magnetically stirred for 1 week at room temperature. The resulting ANF solution showed a dark red color.

**Fabrication of the MXene/nanofiber composite membrane**. The Ti$_3$C$_2$T$_x$ was redispersed in DMSO, and further mixed with certain amount of ANF suspension. The mixture was sonicated for 15 min and stirred for 2 h to form a uniform suspension. The MXene/ANF mixture was vacuum filtered on the anodized aluminum filter film and washed with deionized water for several times. After drying in air for 24 h, the MXene/ANF composite membrane could be easily peeled from the substrate.

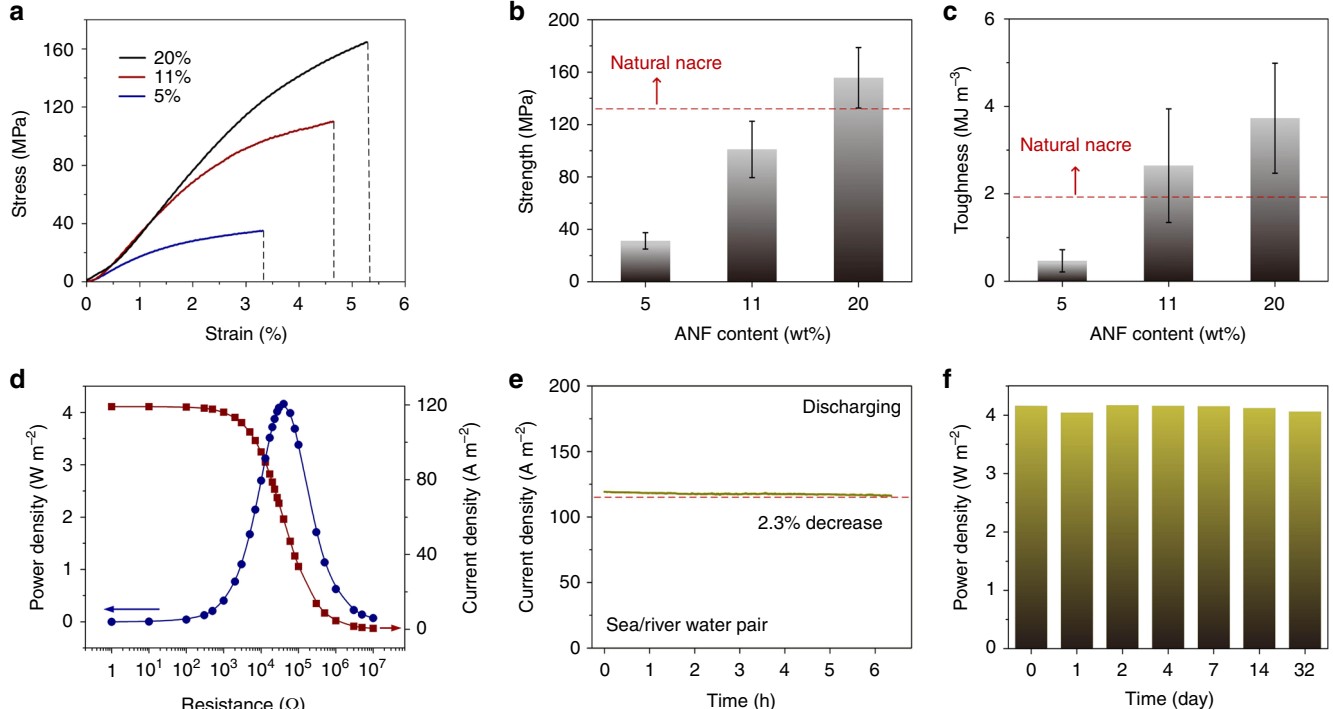

**Fig. 6** Application viability of the MXene/nanofiber composite membrane. **a** Tensile stress–strain curves of composite membranes with different ANF contents. **b**, **c** The ultimate strength (**b**) and toughness (**c**) of the composite membrane with different ANF contents. The dashed red line indicates the corresponding values of natural nacre. Error bars represent s.d. **d** The power output by mixing natural sea water and river water. **e** Current–time curve of system working with no electrolyte replenishing. **f** The stability of the composite membrane-based energy conversion devices under natural water source

**Electrical measurements**. The transmembrane ionic transport and subsequent osmotic energy conversion tests were performed with an electrochemical workstation (CHI). The composite membrane was mounted between a custom-made two-compartment electrochemical cell. The testing membrane area was about $3 \times 10^4 \ \mu m^2$, the same as previous reports[55]. Homemade Ag/AgCl electrodes were used to apply a transmembrane electrical potential and remained stable during the testing process. For the stability test, the membrane was clamped in the electrochemical cell and stayed in the testing solutions all the time, and the testing solutions were replenished before each measurement. The effective salinity of the natural water was calibrated with a standard curve obtained by measuring the conductivity (with a commercial conductivity meter) of a series of standard NaCl with known concentration. The testing solutions were all prepared using ultrapure water (18.2 MΩ cm).

**Characterization**. The SEM image was captured using SEM (Zeiss Gemini 500). The TEM image was captured using JEM 2100F field emission TEM (JEOL, Tokyo, Japan). The XPS was performed in ESCALAB 250Xi. The mechanical characterization of the composite membrane was conducted on a tensile-compressive tester (M5-2) with a loading rate of 1 mm/min. The membrane samples were cut into strips with a width of 2 mm and length of 8 mm. The Young's modulus can be calculated by the slope of the linear region of the stress–strain curves and the toughness was determined by the area under the stress–strain curves. The reported tensile strength, modulus, and toughness were the averages of three samples. The Zeta potential of MXene was performed in solution system using a Zetasizer (Nano ZSP, Malvern Instruments Ltd., Malvern, UK) due to its good dispersibility in water. For the ANF, its DMSO dispersion is cast onto a silicon wafer and then immersed in water to form a free-standing nanofiber membrane. Zeta potential of the membrane system was measured by SurPASS Electro-kinetic Analyzer (Anton-Paar).

**Numerical simulations**. The theoretical calculation is carried out using coupled PNP equations within the commercial finite element package COMSOL script environment (Supplementary Note 3). In order to gain affordable computation scale, we use a 2D model and the fluidic pathway is simplified to be a 1000 nm long single channel (Supplementary Fig. 13). As the resistance of mass transfer at the entrance and exit could significantly influence the transport of ions across the channel, two electrolyte reservoirs (400 × 200 nm) were added. The concentration gradient is set to 50-fold (i.e. 0.5 M/0.01 M). The open-circuit voltage and short-circuit current can be derived from the intercept on X/Y-axis of calculated current–voltage curves. The corresponding power output and energy conversion efficiency can be obtained.

## Data availability
The authors declare that the data supporting the findings of this study are available within the Article and its Supplementary Information files, and all data are available from the authors on reasonable request.

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

## Acknowledgements

Z.Z. acknowledges the support of the Alexander von Humboldt Foundation. This work was financially supported by the European Union's Horizon 2020 research and innovation program under grant agreement No. 785219, the European Science Foundation (ESF), and the Coordination Networks: Building Blocks for Functional Systems (SPP1928, COORNET). The authors acknowledge the Dresden Center for Nanoanalysis (DCN) at TU Dresden.

## Author contributions

Z.Z. and X.F. conceived and designed the experiments and wrote the paper; Z.Z. and S.Y. synthesized the Ti$_3$C$_2$T$_x$ flakes. P.Z. and G.C. did the SEM characterization. Z.Z. carried out the ion transport recording, osmotic energy conversion measurement, numerical calculation, and mechanical testing. P.Z., J.Z., and G.C. rendered helpful discussions. All authors commented on the manuscript.

## Additional information

**Competing interests:** The authors declare no competing interests.

