## [Peer Review File · Nature Communications]

Reviewers' comments:

Reviewer #1 (Remarks to the Author):

The manuscript written by Feng and coworkers reported a fascinating composite material for osmotic energy harvesting with high mechanical performance and high output power. By using MXene and Kevlar nanofibers, a 2D "brick-and-mortar" structure is formed and shows higher mechanical performance than natural nacre. Also, the power density of 4.1 W/m² is a big increase than those of the reported 2D materials. The theoretical analysis well supports the authors' hypothesis. The data are solid and the figures are good demonstrations. This work is interesting and important in the blue energy conversion field. I would be happy to recommend the acceptance of the manuscript after the authors comment on the following concerns.

1. With respect to the calculation of the diffusion potential E_{diff} in Figure 2e, how to get the E_{redox} values? Are E_{redox} values calculated or measured? and what is the difference between the values from these two methods? These data will be a great help for the readers' deep understanding the high energy conversion performance by using this amazing material.
2. The authors have tested the effect of the increasing contents of ANF on the osmotic energy conversion behavior. As stated "In this respect, the excess ANF can partially block the 2D channel and introduce large physical steric hindrance for the ion transport, resulting in decreased ion flux and thus largely weakened power output", is there any experimental support for these words?
3. For the output power density demonstration, the testing area of the membrane is important for the potential applications. What is the testing membrane area and the effect area for the energy conversion? Also, the power data should be provided.
4. The authors have demonstrated the long term stability of the MXene/ANF composite membrane on the energy conversion aspect. It is better to show the testing conditions to support the excellent performance of the amazing materials. Besides, does the membrane stay in the testing solutions all the time? As the mechanical strength is important for the long term application, how about the stability of the membrane in these characterization?
5. The circles in Figure 2d are not mentioned in the figure caption. The Figure 4a should be improved, for example, the different symbols for the two groups of data. The missed error bar for the zeta potential testing (Figure 4d) and stability testing (Figure 6f) should be added. Figure 6e needs improved for better presenting.
6. To help readers understanding this important work, some important paper should be included in the introduction part, Nano Energy, 59, 354; Angew Chem 2015, 54, 3387; AM 2010, 22, 1021. Also, in the energy conversion experiments, the natural sea water/river water from Mediterranean Sea and Elbe River are used, and the salinity of these samples should be provided.

Reviewer #2 (Remarks to the Author):

The manuscript by Zhang et al. reports osmotic power generation using MXene/Kevlar nanofiber composite membrane. The osmotic power generation using two-dimensional (2D) nanochannels and nanopores were previously studied; but, the authors claim that the generated power using Mxene/Kevlar composite outperform other 2D nanochannels based devices. Overall, the manuscript is interesting and well presented. The authors need to provide further technical details to fully assess the validity of the key finding (higher power density generation) of this paper.

1. What is the electrical conductivity of MXene/Kevlar nanofiber membrane and how that influence the ionic transport measurement (e.g., is there any electrochemical reaction on the membrane due to its conductivity?).
2. The area and exact thickness of the membrane used for the study is not provided.
3. How the power density varies with the thickness of the membrane? Is there any critical thickness where power density is maximum?
4. Did the authors calculate the internal resistance of the membrane?
5. Why the authors not compared a pristine Mxene membrane (without fiber) with MXene/Kevlar nanofiber composite membrane? Even though the author's changed the wt.% of fiber, the data from the pristine sample is essential to understand the influence of the fiber on the performance.
6. Why the power density becomes zero for load resistance less than 100 ohms? On the other hand, the current density is very high. Without knowing the area of the sample, these numbers are

meaningless.

7. The reported diffusion current of < 0.3 microAmp is not consistent with $\sim 100\text{A/m}^2$ current density reported in Fig. 3b (otherwise area of the sample should be extremely low, I do not think that is the case after noticing fig. 1f scale bar).

8. The authors explained the performance of the membrane by using space charge mechanism. How to justify the existence of the space charge on an electrically conducting membrane?

9. Technical details/methodology of zeta potential measurements and mechanical strength measurement are missing.

Responses to the comments of the reviewers

Reviewer #1 (Remarks to the Author):

The manuscript written by Feng and coworkers reported a fascinating composite material for osmotic energy harvesting with high mechanical performance and high output power. By using MXene and Kevlar nanofibers, a 2D “brick-and-mortar” structure is formed and shows higher mechanical performance than natural nacre. Also, the power density of 4.1 W/m² is a big increase than those of the reported 2D materials. The theoretical analysis well supports the authors’ hypothesis. The data are solid and the figures are good demonstrations. This work is interesting and important in the blue energy conversion field. I would be happy to recommend the acceptance of the manuscript after the authors comment on the following concerns.

1. With respect to the calculation of the diffusion potential E_{diff} in Figure 2e, how to get the E_{redox} values? Are E_{redox} values calculated or measured? and what is the difference between the values from these two methods? These data will be a great help for the readers’ deep understanding the high energy conversion performance by using this amazing material.

Response: Thank you for your valuable comments. In this work, the value of E_{redox} was measured using an experimental method. The separator membrane was replaced by a nonselective silicon membrane containing a single micro-window. In this case, the measured potential was contributed solely by the asymmetric redox reactions on the electrodes (E_{redox}). The electrode potential remained stable during the calibration process as the diffusion of ions did not affect the bulk concentration obviously in the first several minutes. Such an experimental method can largely preclude the influence brought by many unexpected factors such as the contamination and electrode imperfection (*J. Am. Chem. Soc.* 2014, 136, 12265; *Lab Chip*, 2015, 15, 168; *Nanotechnology* 2013, 24, 345401; *Nano Energy* 2018, 53, 643). In the revised manuscript, we added some description and listed their values in Table S2.

	10-fold	50-fold	100-fold	250-fold	1000-fold
E_{mea} (mV)	34.06	91	122.6	158.36	203.8
E_{redox} (mV)	22.2	48.8	66	76.4	103
E_{diff} (mV)	11.86	42.2	56.6	81.96	100.8

Supplementary Table S2. List of E_{mea} , E_{redox} , and E_{diff} (the low concentration side is set to be 0.1 mM KCl).

2. The authors have tested the effect of the increasing contents of ANF on the osmotic energy conversion behavior. As stated “In this respect, the excess ANF can partially block the 2D channel and introduce large physical steric hindrance for the ion transport, resulting in decreased ion flux and thus largely weakened power output”, is there any experimental support for these words?

Response: Thank you for your comments. Although the ANF can bring mobile charge to the confined interlayer spacing between MXene nanosheets, it can also introduce non-negligible physical blockage due to its relatively large size which will decrease the effective transporting route for the ions and undermine the transport efficiency. Such blocking effect has also been observed in other systems such as polymer-nanofiber modified stimuli-responsive nanochannels (*Small* 2009, 11, 1287; *Adv. Mater.* 2010, 22, 2440). As a consequence, the measured power density of the MXene/ANF composite system drops substantially from about 3.7 W/m² to 1.9 W/m² as the ANF content increases from 11 % to 30% (Fig. 3c).

3. For the output power density demonstration, the testing area of the membrane is important for the potential applications. What is the testing membrane area and the effect area for the energy conversion? Also, the power data should be provided.

Response: Thank you for your constructive comments. We apologize for missing the testing area of the sample. The testing membrane area (*i.e.* effective area) is about $3 \times 10^4 \mu\text{m}^2$, the same as the previous reports (for example, 1D mesoporous carbon membrane, 3.46 W/m², *J. Am. Chem. Soc.* 2014, 136, 12265; 2D graphene oxide membrane, 0.76 W/m², *ACS Nano* 2017, 11, 10816; 3D polymer/MOF composite network, 2.87 W/m², *Nano Energy* 2018, 53, 643). The use of small working area may help to reduce the influence of the “*hindered diffusion transport of counter-ion due to increased number of co-ion in nanopore networks upon increasing working area*” (*Desalination* 2018, 425, 156). Through this way, the measured energy conversion property will be closer to the initial nature of the power generation system (*Nature* 2013, 494, 455). The power data are provided in Fig. 3 and Fig. 6, and are both normalized to the effective area.

4. The authors have demonstrated the long term stability of the MXene/ANF composite membrane on the energy conversion aspect. It is better to show the testing conditions to support the excellent performance of the amazing materials. Besides, does the membrane stay in the testing solutions all the time? As the mechanical strength is important for the long term application, how about the stability of the membrane in these characterization?

Response: Thank you for your suggestions. During the stability testing, the membrane was clamped in the electrochemical cell and stayed in the testing solutions all the time, and the testing solutions were replenished before each measurement. After long-term testing, the membrane was still very robust and there were no observable morphology changes, indicating its good stability. For clarity, we added some description in the main text and experimental section.

5. The circles in Figure 2d are not mentioned in the figure caption. The Figure 4a should be improved, for example, the different symbols for the two groups of data. The missed error bar for the zeta potential testing (Figure 4d) and stability testing (Figure 6f) should be added. Figure 6e needs improved for better presenting.

Response: Thank you for your comments. Figure 4a, Figure 4d, and Figure 6e are improved in the revised manuscript. We also added relevant description of Figure 2d in the main text. For the stability testing, the composite membrane was very stable under each measurement, so we did not add error bar in Figure 6f.

Figure R1. (a) Revised Figure 4a. (b) Revised Figure 4d. (c) Revised Figure 6e.

6. To help readers understanding this important work, some important paper should be included in the introduction part, Nano Energy, 59, 354; Angew Chem 2015, 54, 3387; AM 2010, 22, 1021. Also, in the energy conversion experiments, the natural sea water/river water from Mediterranean Sea and Elbe River are used, and the salinity of these samples should be provided.

Response: Thank you for your comments. The mentioned important papers have been cited in the revised manuscript. The salinity of the natural water was calibrated with a standard curve obtained by measuring the conductivity (with a commercial conductivity meter) of a series of standard NaCl with known concentration. The effective salinity degree of the water from Mediterranean Sea and Elbe River are 0.6 M and 0.004 M NaCl, respectively.

Reviewer #2 (Remarks to the Author):

The manuscript by Zhang et al. reports osmotic power generation using MXene/Kevlar nanofiber composite membrane. The osmotic power generation using two-dimensional (2D) nanochannels and nanopores were previously studied; but, the authors claim that the generated power using Mxene/Kevlar composite outperform other 2D nanochannels based devices. Overall, the manuscript is interesting and well presented. The authors need to provide further technical details to fully assess the validity of the key finding (higher power density generation) of this paper.

1. What is the electrical conductivity of MXene/Kevlar nanofiber membrane and how that influence the ionic transport measurement (e.g., is there any electrochemical reaction on the membrane due to its conductivity?).

Response: Thank you for your valuable comments. The electrical conductivity of the optimized MXene/Kevlar nanofiber membrane (ANF content: 11 %) is approximately $36.7 \pm 7 \text{ k}\Omega/\text{sq}^{-1}$

(corresponding to a conductivity $\sim 10 \text{ S m}^{-1}$), similar to the previous reported MXene based composites (e.g. *Sci. Adv.* 2018, 4, eaaq0118). The sheet resistance value was higher than that of pure MXene sheets because of the presence of insulating ANF. Here we would like to emphasize that the electrical conductivity of the membrane will not influence the transmembrane ion transport. In our experiment, two parts of electrolyte solution are separated by the MXene/ANF composite membrane. Due to the existence of the mobile surface charge in the MXene and ANF, electric double layers (EDLs) will form between the interlayer nanochannels. The EDLs will effectively repel the co-ions, allowing the counter-ions to transport through. To balance the surface charge, the counter-ions will be concentrated inside the channels, thus enhancing the ionic conductivity (*Nat. Nanotechnol.* 2009, 4, 713; *Rev. Mod. Phys.* 2008, 80, 839). The composite membrane actually acts as an ion transporting media and thus the energy conversion process relies on the flow of ions. The electrical conductivity of a material will influence the transport of electrons rather than ions. Therefore, nanofluidic power generation requires that material should exhibit high surface charge density rather than high electrical conductivity. However, the electrical conductivity can also be utilized in nanofluidic applications. For example, one can apply an external potential onto the membrane to regulate the surface potential of the confined channel, enabling electrostatic modulation of ion diffusion through the 2D membrane (*Nat. Nanotechnol.* 2018, 13, 685) or membrane based field effect ionic transistor (*Appl. Phys. Lett.* 2013, 102, 213108).

2. The area and exact thickness of the membrane used for the study is not provided.

Response: We apologize for missing these parameters in the previous submission. The testing membrane area is about $3 \times 10^4 \mu\text{m}^2$, the same as the previous reports (for example, 1D mesoporous carbon membrane, 3.46 W/m^2 , *J. Am. Chem. Soc.* 2014, 136, 12265; 2D graphene oxide membrane, 0.76 W/m^2 , *ACS Nano* 2017, 11, 10816; 3D polymer/MOF composite network, 2.87 W/m^2 , *Nano Energy* 2018, 53, 643). The use of small working area may help to reduce the influence of the “*hindered diffusion transport of counter-ion due to increased number of co-ion in nanopore networks upon increasing working area*” (*Desalination* 2018, 425, 156). Through this way, the measured energy conversion property will be closer to the initial nature of the power generation system (*Nature* 2013, 494, 455). Additionally, the thickness of the membrane for electrical measurement is approximately $4.5 \mu\text{m}$ and the thickness can be adjusted through the dosage of the colloidal dispersion used for filtration. According to your suggestion, we also investigated the influence of the thickness on the energy conversion performance. Please see the next comment.

3. How the power density varies with the thickness of the membrane? Is there any critical thickness where power density is maximum?

Response: Thank you for your constructive suggestions. In the revised manuscript, we systematically investigated the influence of the thickness on the energy conversion performance. As shown in Figure S10, the power density scales inversely with the thickness, which can be ascribed to the decreased fluid resistance upon increasing the membrane thickness.

In principle, there will be a critical thickness where the power density is maximum. If the membrane is too thin, the ion selectivity will be undermined and there will be also strong ion concentration polarization, particularly at the low-concentration side, leading to weakened energy conversion performance. Such a critical thickness is predicted to be in the sub-micrometre scale. However, in our experiment, it was difficult to peel off a free-standing membrane below 2 μm and thus we did not measure such critical values.

Supplementary Figure S10. Influence of the thickness of the composite membrane (ANF content: 11 %) on the power density.

4. Did the authors calculate the internal resistance of the membrane?

Response: Thank you for your comments. The internal resistance of the membrane can be calculated as the ratio of the open-circuit voltage and short-circuit current. Under the concentration gradient of artificial river water and sea water (0.5 M/0.01 M), the calculated inner resistance from the current-voltage curve (Figure S9) is about 29 $\text{k}\Omega$. Notably, under the same condition, the output power density achieves a maximum value at a resistance of about 27 $\text{k}\Omega$ which is close to this value. If a nanofluidic power source is consistent with normal voltaic batteries, the maximum power output is commonly achieved with a resistance equal to the internal resistance of the membrane. This discussion has been added in the revised manuscript.

5. Why the authors not compared a pristine MXene membrane (without fiber) with MXene/Kevlar nanofiber composite membrane? Even though the author's changed the wt.% of fiber, the data from the pristine sample is essential to understand the influence of the fiber on the performance.

Response: Thank you for your comments. Actually, we did not test the power density of the pristine MXene as we could not obtain a free-standing membrane for such a thin thickness ($\sim 5 \mu\text{m}$). While the power output of pristine and composite samples can also be compared in parallel through adopting non-functional porous membrane as the supporting layer. In the revised manuscript, we measured the power density of pristine MXene and MXene/ANF 11%

membrane both supported by non-functional porous hydrophilic nylon-66 membranes (pore size~0.2 μm). As shown in Figure S11, the power density of composite system (2.74 W/m^2) is much larger than that of the pristine system (1.44 W/m^2), indicating the enhancing effect of the nanofiber on the energy conversion performance. Notably, the power density is lower than that of unsupported samples as the supporting membrane will introduce extra fluid resistance.

Supplementary Figure S11. Comparison of the power density of pristine MXene and MXene/ANF composite membranes. The power density of composite system (2.74 W/m^2) is much larger than that of the pristine system (1.44 W/m^2), indicating the enhancing effect of the nanofiber on the energy conversion performance. Here, the pristine MXene and MXene/ANF (11%) are both supported by a non-functional porous hydrophilic nylon-66 membranes (pore size~0.2 μm).

6. Why the power density becomes zero for load resistance less than 100 ohms? On the other hand, the current density is very high. Without knowing the area of the sample, these numbers are meaningless.

Response: Thank you for your comments. As shown in *the response of Comment 2*, the testing membrane area is about $3 \times 10^4 \mu\text{m}^2$. The power density becomes zero for load resistance less than 100 ohms. Actually, the harvested osmotic energy is output to external circuit to supply an external load. The power density of the external load can be calculated as $P = I^2 \times R_L$, where I is the current flowing through the circuit and R_L is the load resistance. Although there will be high current density at low load resistance, the overall results obtained from the equation is very low.

7. The reported diffusion current of < 0.3 microAmp is not consistent with ~ 100A/m² current density reported in Fig. 3b (otherwise area of the sample should be extremely low, I do not think that is the case after noticing fig. 1f scale bar).

Response: Thank you for your comments. As shown in *the response of Comment 2*, the testing membrane area is about $3 \times 10^4 \mu\text{m}^2$. The diffusion current is not normalized to the effective membrane area, making it inconsistent with ~ 100A/m² current density (which has been

normalized). In the revised manuscript, to avoid misleading, the unit of diffusion current is changed into the same unit.

Figure 2e. The recorded diffusion potential and diffusion current under a series of concentration gradient.

8. The authors explained the performance of the membrane by using space charge mechanism. How to justify the existence of the space charge on an electrically conducting membrane?

Response: Thank you for your comments. In this work, the ANFs are negatively charged in electrolyte solution and their entanglement in confined channel can create a negatively charged space charge zone. Here we would like to clarify that the space charge is a cluster of surface charge (contributed by many surface-charged ANFs). Such type of surface charge cluster can also be found in polymer-brushes-filled nanochannel and charged polyelectrolyte gel, and the simplification of them using space charge model has been widely reported (*Phys. Chem. Chem. Phys.* 2014, 16, 2465; *J. Am. Chem. Soc.* 2017, 139, 1396). Therefore, the characterization of the existence of the space charge is identical to the characterization of the existence of the surface charge of ANF (as shown in Fig. 4d).

9. Technical details/methodology of zeta potential measurements and mechanical strength measurement are missing.

Response: Thank you for your comments. We apologize for missing these important details in the previous submission. Additionally, we checked the throughout the manuscript and also added other technical details. In detail, The SEM image was captured using scanning electron microscope (Zeiss Gemini 500). The TEM image was captured using JEM 2100F field emission transmission electron microscope (JEOL, Tokyo, Japan). The XPS was performed in ESCALAB 250Xi. The mechanical characterization of the composite membrane was conducted on a tensile-compressive tester (M5-2) with a loading rate of 1 mm/min. The membrane samples were cut into strips with a width of 2 mm and length of 8 mm. The Young's modulus can be calculated by the slope of the linear region of the stress-strain curves and the toughness was determined by the area under the stress-strain curves. The reported tensile

strength, modulus, and toughness were the averages of three samples. The Zeta potential of MXene was performed in solution system using a Zetasizer (Nano ZSP, Malvern Instruments Ltd., Malvern, UK) due to its good dispersibility in water. For the ANF, its DMSO dispersion is cast onto a silicon wafer and then immersed in water to form a free-standing nanofiber membrane. Zeta potential of the membrane system was measured by SurPASS Electro-kinetic Analyzer (Anton-Paar).

REVIEWERS' COMMENTS:

Reviewer #1 (Remarks to the Author):

The authors have well addressed my concerns and the manuscript has been considerably improved. I therefore recommend the acceptance of this manuscript for publication.

Reviewer #2 (Remarks to the Author):

Even though the authors satisfactorily answered all of my questions, I am surprised with the fact that the testing area of the membrane is too low ($3 \times 10^4 \mu\text{m}^2$) compared to the size of the MXene/Kevlar nanofiber composite membranes prepared and demonstrated in Fig.1. As the authors mentioned in the reply, this could be purposefully kept small to compare to the previously published papers. However, the authors also mention that "the use of small working area may help to reduce the influence of the hindered diffusion transport of counter-ion due to increased number of co-ion in nanopore networks upon increasing working area". This implies that the power generated may not increase linearly with the area of the membrane and hence the power normalisation with the area of the membrane may not be logical. This could potentially lead to an overestimation of the power per square meter area reported in this manuscript (the real area used in this manuscript is orders of magnitude smaller). Therefore, the authors need to report the variation of the power generated as a function of the membrane area (1 cm² area seems achievable in these experiments) and the associated discussions in the manuscript.

Responses to the comments of the reviewers

Reviewer #1 (Remarks to the Author):

The authors have well addressed my concerns and the manuscript has been considerably improved. I therefore recommend the acceptance of this manuscript for publication.

Response: We greatly thank the reviewer for the recommendation to accept the paper for publication.

Reviewer #2 (Remarks to the Author):

Even though the authors satisfactorily answered all of my questions, I am surprised with the fact that the testing area of the membrane is too low ($3 \times 10^4 \mu\text{m}^2$) compared to the size of the MXene/Kevlar nanofiber composite membranes prepared and demonstrated in Fig.1. As the authors mentioned in the reply, this could be purposefully kept small to compare to the previously published papers. However, the authors also mention that “the use of small working area may help to reduce the influence of the hindered diffusion transport of counter-ion due to increased number of co-ion in nanopore networks upon increasing working area”. This implies that the power generated may not increase linearly with the area of the membrane and hence the power normalisation with the area of the membrane may not be logical. This could potentially lead to an overestimation of the power per square meter area reported in this manuscript (the real area used in this manuscript is orders of magnitude smaller). Therefore, the authors need to report the variation of the power generated as a function of the membrane area (1 cm² area seems achievable in these experiments) and the associated discussions in the manuscript.

Response: Thank you for your valuable comments. As suggested, the variation of power density upon increasing the membrane area was tested. About 30 % decrease of power density was observed when the area increased by about 7-times (Note that the membrane area is restricted by the fixed testing mold and thus cannot be changed in a wide range), indicating that the generated power did not increase linearly with the area of the membrane, which can be ascribed to the combined effect of multiple factors in multi-scale. For example, as the membrane area increases, the entering resistance (*i.e.* the sum of reservoir resistance and reservoir/nanopores interfacial resistance) will play an increasingly dominant role that largely undermines the power output. At the same time, more and more stochastic physical defects within the membrane network will appear, which will weaken the ion selectivity of the membrane and decrease the osmotic transport ability.

Notably, such unlinear response also exists in many other osmotic energy conversion systems (e.g. *Nano Energy* 2018, 53, 643; *J. Am. Chem. Soc.* 2014, 136, 12265). Actually, there exist gaps between the sing-pore devices and the multi-pores membranes, and also between the small-scale demonstrations and the industrial large-scale applications. While the normalization of power output with the area of the membrane is commonly acceptable and has been adopted by many researchers such as A. Majumdar (USA), L. Jiang (China), A. Radenovic

(Switzerland), and K. Nijmeijer (Netherlands) (e.g. *Microfluid. Nanofluid.* 2010, 9, 1215; *Sci. Adv.* 2018, 4, eaau1665; *Nature* 2016, 536, 197; *Nanotechnology* 2013, 24, 345401). The translation of the normalized high power density into real high power for industrial large-scale membrane applications remains challenging. Understanding the origin and thereby bridging the gap is becoming an important research direction that attracts attentions of both experimental and theoretical scientists. To this end, optimizing the membrane fabrication protocols to prepare defect-free membranes is generally considered an effective strategy. Recently, Gao *et al.* reported a theoretical study to show that several technical improvements (e.g. maximizing the surface charge, optimizing the membrane thickness, and eliminating concentration polarization) can be used to dramatically increase the power density of membrane-scale osmotic power generation by eliminating the entering resistance (*Small* 2019, 15, 1804279). Based on the above discussions, we prefer not to include the study of large-area experiment into the main text and supplementary information.